# Studies on the Volatiles Composition of Stored Sheep Wool, and Attractancy toward *Aedes aegypti* Mosquitoes

**DOI:** 10.3390/insects13020208

**Published:** 2022-02-18

**Authors:** Maia Tsikolia, Nurhayat Tabanca, Daniel L. Kline, Betul Demirci, Liu Yang, Kenneth J. Linthicum, Jeffrey R. Bloomquist, Ulrich R. Bernier

**Affiliations:** 1European Biological Control Laboratory, U.S. Department of Agriculture-Agricultural Research Service (USDA-ARS), 54623 Thessaloniki, Greece; 2Subtropical Horticulture Research Station (SHRS), U.S. Department of Agriculture-Agricultural Research Service (USDA-ARS), Miami, FL 33158, USA; nurhayat.tabanca@usda.gov; 3Center for Medical, Agricultural, and Veterinary Entomology (CMAVE), U.S. Department of Agriculture-Agricultural Research Service (USDA-ARS), Gainesville, FL 32608, USA; dan.kline@usda.gov (D.L.K.); kenneth.linthicum@usda.gov (K.J.L.); 4Department of Pharmacognosy, Faculty of Pharmacy, Anadolu University, Eskisehir 26470, Turkey; bdemirca@anadolu.edu.tr; 5Department of Entomology and Nematology, Emerging Pathogens Institute, University of Florida, Gainesville, FL 32610, USA; lzy0017@tigermail.auburn.edu (L.Y.); jbquist@epi.ufl.edu (J.R.B.)

**Keywords:** GC-FID, GC-MS, hydrodistillation, SPME, HS-Hayesep-Q, semiochemicals

## Abstract

**Simple Summary:**

In order to discover new natural remedies for controlling insect vectors, stored sheep wool was tested for its attractiveness to mosquitoes. The volatile compounds from the wool were collected. A total of 52 compounds were detected; many of them known to have mosquito attractant activity against various species. The most abundant compound in the sheep wool hydrodistillate was not attractive for female adult *Aedes aegypti* mosquitoes in small-scale lab evaluations. Comparative evaluations of raw sheep wool in two semi-field, large, screened outdoor cages each with bait equipped with a U.S. Centers for Disease Control (CDC) trap, revealed the bait with vibrated sheep wool was the best attractant for female adult *Ae. aegypti* mosquitoes compared to the setups without wool or not vibrated. It was concluded that sheep wool, an easily available, affordable, and environment-friendly material, could be considered as a potential tool to be used in dynamic bait setups for mosquito management and surveillance.

**Abstract:**

To discover new natural materials for insect management, commercially available stored sheep wool was investigated for attractancy to female adult *Aedes aegypti* mosquitoes. The volatiles from sheep wool were collected by various techniques of headspace (HS) extractions and hydrodistillation. These extracts were analyzed using gas chromatography–mass spectrometry (GC-MS) and gas chromatography–flame ionization detector (GC-FID) coupled with GC-MS. Fifty-two volatile compounds were detected, many of them known for their mosquito attractant activity. Seven compounds were not previously reported in sheep products. The volatile composition of the extracts varied significantly across collections, depending on the extraction techniques or types of fibers applied. Two types of bioassay were conducted to study attractancy of the sheep wool volatiles to mosquitoes: laboratory bioassays using glass tubes, and semi-field bioassays using large, screened outdoor cages. In bioassays with glass tubes, the sheep wool hydrodistillate and its main component, thialdine, did not show any significant attractant activity against female adult *Ae. aegypti* mosquitoes. Semi-field bioassays in two large screened outdoor cages, each equipped with a U.S. Centers for Disease Control (CDC) trap and the various bait setups with Vortex apparatus, revealed that vibrating wool improved mosquito catches compared to the setups without wool or with wool but not vibrating. Sheep wool, when vibrated, may release intensively volatile compounds, which could serve as olfactory cues, and play significant role in making the bait attractive to mosquitoes. Sheep wool is a readily available, affordable, and environment-friendly material. It should have the potential to be used as a mosquito management and surveillance component in dynamic bait setups.

## 1. Introduction

In this study, we aimed to investigate natural means for spatial protection against mosquito bites and/or find new tools for surveillance purposes. Vector-borne diseases such as malaria, lymphatic filariases, schistosomiasis, leishmaniasis, onchocerciasis, trypanosomiasis, dengue, yellow fever and Japanese encephalitis, produce significant morbidity and mortality in humans and livestock, mostly in the developing world [1]. *Aedes aegypti* (Diptera: Culicidae) is a primary vector, which transmit dengue, as well as, yellow fever, chikungunya, and Zika [2]. Dengue, sometimes causing a potentially lethal complication called severe dengue, is considered the most important mosquito-borne viral disease in the world today with about half of the world’s population now at risk [3]. Vector control using pesticides plays an important role in preventing vector-borne disease. Synthetic pesticides are the major and important tools in controlling vectors and, accordingly, preventing diseases transmission. Although insect populations develop resistance to certain synthetic pesticides, they can have adverse effects on non-targeted species, and natural materials have been considered as potential tools for pest management [4]. Search for mosquito-attractive semiochemicals has been a subject of interest in the last few decades. New compounds or odor blends are regularly proposed as lures for odor-baited traps [5].

There are a few reports in the literature where sheep were used as bait for rift valley fever virus (RVFV) vectors [6,7,8]. Studies by Tchouassi et al. [7,8] found that including sheep and other animal odors increases light trap catch of RVF virus mosquito vectors with U.S. Centers for Disease Control (CDC) light or CO_2_; and the odor may be the primary factor for mosquito attraction. According to Tchouassi et al. [8], aldehydes: heptanal, octanal, nonanal, decanal, present in RVF virus hosts’ odors, including sheep, could play an important role in the attractancy of RVF mosquitoes. Yan at al. [9] studied the volatiles from Merino sheep wool samples collected by headspace SPME, and analyzed these by GC–MS–electroantennography; the samples containing octanal and nonanal possessed attractiveness to *Lucilia cuprina* (Diptera: Calliphoridae) flies.

The chemical composition of collected volatiles can be affected by various factors; e.g., Vasta et al. [10] using solid-phase microextraction (SPME) and GC–MS analysis of lamb fat samples, detected 6-methyl-2-heptanone and 2,5-dimethyl-4-hydroxy-3(2H)-furanone that were related to the time of grazing. Almela et al. [11] using SPME with fiber coating—divinylbenzene-carboxen-polydimethylsiloxane and GC–MS methods, investigated aldehydes, alcohols, ketones, phenols, indole, and sulfur compounds, and found that the ewe’s diet strongly affected the volatile compounds profile of the cooked meat. According to Burger et al. [12] bioassays during the lambing seasons confirmed the role of lamb odors in ewe–lamb recognition; they identified 133 volatile organic compounds in cranial wool of Döhne Merino lambs and found that the wool volatiles of twins are practically identical, but differ from those of other twins or non-twin lambs in the flock. The analysis of the headspace SPME volatile collection of a variety of scoured wool samples and greases using a carboxen-polydimethylsiloxane coated fiber, by a GC–MS and GC/pulsed flame photometric detector, showed the number of volatiles decreased after various stages of wool scouring [13].

Sheep wool is an agricultural waste produced by sheep breeding; it is a renewable, recyclable and environmentally friendly material, used in many fields and sectors, offering significant benefits for sustainability [14]. Wool fiber decomposition does not pose a risk to the environment and may even be used as a plant fertilizer [15]. Along with other advantages, sheep wool is attractive as an odor source because of its low cost.

In this work, we studied commercially available, stored sheep wool volatiles using various extraction and analysis techniques/methods, to identify the molecular composition of sheep wool odor, as a potential attractant for mosquitoes, and conducted bioassays to investigate the possibility of using this readily available and inexpensive material for mosquito management and surveillance. The molecular composition of sheep wool volatiles and, correspondingly, the potential activity against insects, could vary depending on the breed, species, age, diet, location, etc. Evaluation of the feasibility and reliability of this material, as a source of the desired volatiles, should be assessed taking into account all the variabilities, and is out of the scope of this work.

## 2. Materials and Methods

### 2.1. Chemicals

Ionol (Cas #128-37-0, 2,6-di-tert-butyl-4-methylphenol (BHT)) was purchased from Sigma-Aldrich Co., (St. Louis, MO, USA). The thialdine (5,6-dihydro-2,4,6-trimethyl-4*H*-1,3,5-dithiazine) was purchased from Enamine LLC (Monmouth Jct., Middlesex County, NJ, USA). Raw sheep wool (100%), purchased via Amazon from Flowing River farm (Arcadia, WI, USA), was unwashed and unsorted, 4–5 inch staple length, from Dorset Horn, North Country Cheviot and Île-de-France sheep, shorn at the farm in West Central Wisconsin. The sheep were raised in Wisconsin, had seasonal free range grazing, and were fed a well-balanced diet of assorted grains and forages.

### 2.2. Extraction of Volatile Compounds

#### 2.2.1. Hydrodistillation

Sheep wool (160 g) was placed in a 3 L round bottom flask and water (~2000 mL) was added to cover the wool material, and water (~2 mL) was also added to the Clevenger unit (Figure 1). The distillation system was covered with aluminum foil and the temperature set to 100 °C and the water-cooling system set on high stream. After 1 h of heating, the mixture in the flask started to boil and distillation started. The system was allowed to distill for 3 h, and then heating was stopped to let the system cool [16,17]. Then, hydrosol (5 mL) was subjected to triplicate extractions with 2 mL of *n*-hexane. The condenser was washed with *n*-hexane (2–3 mL) and this hexane solution also was collected. Both solutions were combined, dried over anhydrous sulfate (Na_2_SO_4_) and stored at 4 °C before analysis.

#### 2.2.2. Dynamic Headspace Collection (DHS) by Hayesep-Q Polymer Adsorbent

Sheep wool (100 g) was placed in a polyester Reynolds oven bag (482 mm × 596 mm, Reynolds consumer products, Lake Forest, IL, USA), which prior to use was baked in the oven for 10 h at 80 °C. Volatiles were collected using the Hayesep-Q polymer (0.5 g) adsorbent (60–80 mesh, Hayes Separations, Inc.; Bandera, TX, USA) packed in a glass volatiles collection tube [18]. Prior to the experiment this collection tube was washed with hexane; the purity of this hexane-wash was verified by GC–MS, and the tube was baked in the oven at 50–60 °C, for 5 h. Then this tube was placed in a volatiles collection system (Figure 2) equipped with charcoal-filtered pressurized air entering the sealed container at 0.6 L/min. An outgoing tube from the oven bag with wool led to the volatile collection tube, and a vacuum pump with a vacuum gauge pulled air at 0.6 L/min through the Hayesep-Q adsorbent material. Oven bag openings were sealed with black steel fold-back binder clips (medium, Business Source Products, Post Falls, ID, USA). Pressurized air and vacuum tubes (Figure 2) of corrugated fluorinated ethylene propylene (FEP) having 6.4 × 5.6 × 0.4 mm diameters and wall thickness, respectively, of 1.5 m length (Cole-Parmer North America, Vernon Hills, IL, USA) were fixed with Swagelok brass fittings (Swagelok, Jacksonville, FL, USA). The collection of volatiles continued for 6 h, the collection tube was removed and then eluted with 200 μL of hexane into a glass vial, and it was stored at 4 °C before analysis.

#### 2.2.3. Headspace (HS) Collection by Solid-Phase Microextraction (HS-SPME)

A manual SPME device (Supelco, Bellafonte, PA, USA) using different fibers composed of polydimethylsiloxane/divinylbenzene (PDMS/DVB), carboxen-polydimethylsiloxane (CAR/PDMS), and polydimethylsiloxane (PDMS) (Supelco Inc., Bellefonte, PA, USA) was used for extraction of the sheep wool volatiles [19,20]. Sheep wool (0.2 mg) was transferred to a 15 mL vial and sealed with parafilm [21]. The extraction fiber was pushed through the parafilm layer for exposure to the headspace of the extract for 15 min at 40 °C. The blank collection with the parafilm was performed, as well, in the same conditions. The fiber was then inserted immediately into the injection port of the GC–MS for desorption of the adsorbed volatile compounds for analysis.

### 2.3. Mass Spectrometry Analysis

#### 2.3.1. Gas Chromatography–Mass Spectrometry (GC–MS) Analysis with Closed Electron Ionization (CEI)

A 1 μL aliquot from each extract was analyzed along with hexane blanks using a ThermoFinnigan DSQ (Thermo Fisher Scientific; Austin, TX, USA) gas chromatograph (GC) equipped with a DB-5 column (Agilent; Santa Clara, CA, USA; 30 m × 0.25 mm ID; 0.25 μm film thickness). The GC oven temperature program was set at an initial temperature of 50 °C and held at that temperature for 10 min post-injection, ramped at 2 °C/min to 180 °C, and then ramped at 10 °C/min to 240 °C. The mass analyzer was scanned at a rate of 0.5 sec over a mass range of *m/z* 35 to 650. The programmed temperature vaporizing (PTV) injection port was operated at 200 °C in split mode, the transfer line was set to 200 °C, and the carrier gas was set to a constant flow of 1.5 mL/min. MS were recorded at 70 eV. The results of analysis are shown in Table 1.

#### 2.3.2. GC–Flame Ionization Detector (FID) and GC–MS with Electron Impact (EI) Ion Source

GC analysis was carried out using an Agilent 6890N GC system (SEM Ltd., Istanbul, Turkey) with the FID detector temperature set at 300 °C. To obtain the same elution order with GC–MS, a simultaneous injection was undertaken using the same column and the same operational conditions. Relative percentage amounts of the separated compounds were calculated from FID chromatograms. The results of analysis are shown in Table 2.

The GC–MS analysis was carried out with an Agilent 5975 GC-MSD system (SEM Ltd., Istanbul, Turkey). An Innowax FSC column (60 m × 0.25 mm, 0.25 µm film thickness, SEM Ltd., Istanbul, Turkey) was used with helium as carrier gas (0.8 mL/min). The GC oven temperature program was set at an initial temperature of 60 °C and held at that temperature for 10 min post-injection, ramped at 4 °C/min to 220 °C, held at that temperature for 10 min, and ramped at 1 °C/min to 240 °C. The injector temperature was 250 °C. Mass spectra were recorded at 70 eV, where the mass range was from *m/z* 35 to 450.

Compound Identification: Identification of the volatile components was carried out by comparison of their relative retention times with those of authentic samples or by comparison of their relative retention indices (RRI)/Kovats retention indices, to a series of *n*-alkanes [22]; compounds were analyzed under the same conditions, and compared with literature data. Non-isothermal Kovats retention indices were calculated using equation: *RRI_x_* = 100 *n* + 100 (*t_x_ − t_n_*)/(*t_n+1_ − t_n_*), where t_n_ and t_n+1_ are retention times of the reference *n*-alkanes eluting before and after compound “X”; t_x_ is the retention time of compound “X” [22]. Comparisons of MS fragmentation patterns with those of standards and a mass spectrum database search were performed using the Wiley GC–MS library (Wiley, New York, NY, USA), and the NIST library (US National Institute of Standards and Technology, Gaithersburg, MD, USA). Computer matching against commercial libraries (Wiley GC–MS Library, MassFinder 4 Library) [23,24] and in-house “Baser Library of Essential Oil Constituents” built up by genuine compounds and components of known oils (when identified by GC–FID/GC–MS), as well as MS literature data [25] were also used for the identification.

### 2.4. Nuclear Magnetic Resonance (NMR) Analysis

NMR analyses were conducted at the Center for Nuclear Magnetic Resonance Spectroscopy, University of Florida, Gainesville, FL, USA. NMR spectra were recorded in deuterated chloroform (CDCl_3_, Sigma-Aldrich, St. Louis, MO, USA) with TMS (tetramethylsilane) as the internal standard for ^1^H (500 MHz) and CDCl_3_ as the internal standard for ^13^C (125 MHz). Hydrodistillate was subjected to the ^1^H and ^13^C NMR analysis and compared to neat thialdine analysis results. The NMR shifts of a major compound in hydrodistillate and thialdine were identical.

Sheep hair hydrodistillate: off-white oil, ^1^H NMR (CDCl_3_) *δ* 4.2 (q, *J =* 6.9 Hz, 1H), 4.07 (q, *J =* 6.9 Hz, 1H), 1.48 (d, *J =* 7.0 Hz, 3H), 1.45 (d, *J =* 6.5 Hz, 6H). ^13^C NMR (CDCl_3_) *δ* 61.2, 44.1, 22.7, 21.9.

### 2.5. Bioassays

The mosquito species used for testing was *Ae. aegypti* (Orlando strain, 1952) from colonies maintained at the Mosquito and Fly Research Unit at the US Department of Agriculture–Agricultural Research Service, Center for Medical, Agricultural, and Veterinary Entomology (USDA-ARS CMAVE) in Gainesville, FL, USA. Newly emerged mosquitoes were maintained on 10% sugar water and kept in laboratory cages at an ambient temperature of 28 ± 1 °C and relative humidity (RH) of 35–60%. To investigate attractancy against adult female *Ae. aegypti* mosquitoes, raw sheep wool, the sheep wool hydrodistillate, and thialdine (the major component of hydrodistillate) were tested in the laboratory assays using glass tubes, and in the semi-field assays using the large screened outdoor cages. We could not perform bioassays with HS extracts because of their small quantities.

#### 2.5.1. Glass Tube Attractancy Assays for Thialdine and Sheep Wool Hydrodistillate

The sheep wool hydrodistillate and its major component, thialdine, were subjected to the attractancy assays against *Ae. aegypti* mosquitoes using the glass tubes (Figure 3, dimensions: 2.5 cm diameter and 12.5 cm length, TriKinetics Inc. Waltham, MA, USA), with two treatment caps (Corning Inc., New York, NY, USA) each of which held one filter paper (2.5 cm diameter, Sigma-Aldrich, St. Louis, MO, USA) [26]. Netting (purchased from http://www.michaels.com (accessed on 5 August 2018) was put between the end of glass tube and treated caps to prevent contact of treated paper by the mosquitoes. The glass tube sets were held vertically in a foam board made for the 50 mL centrifuge tubes. At the bottom cap, the filter paper was treated with 50 µL of tested chemicals in acetone solution. The top cap was treated with 50 µL of acetone as control. The tested chemicals were diluted at 0.01, 0.1, 1, and 10 µg/cm^2^ concentrations. The treated filter paper was allowed 10 min to evaporate the solvent before inserting into the caps. Sixteen 3- to 7-day-old female mosquitoes, ice anesthetized, were induced into the glass tube, and allowed 15 min to recover before testing. A line was drawn in the middle of the tube and the proportion of mosquitoes on the chemical-treated side was recorded after 15, 30 and 60 min. The positive control was 1 µg/cm^2^ of the attractant 1-octen-3-ol [27] and acetone treated paper on both caps was tested as solvent control. Attractancy was scored as a fraction of the mosquitoes on the chemically treated side at 15, 30, and 60 min. Thus, an initial score with eight mosquitoes on either side of the midline is 0.5 and attractancy is expressed as a score > 0.5 to 1.0 [26]. Each treatment was replicated three times. Co-applications of wool hydrodistillate at selected concentrations (0.1, 0.01 µg/cm^2^) and 10 mL of CO_2_ were also tested and replicated three times to determine whether CO_2_ would enhance the attractancy to *Ae. aegypti*.

#### 2.5.2. Semi-Field Attractancy Assays with Sheep Wool Using Centers for Disease Control (CDC) Traps

Two hundred nulliparous 6- to 8-day-old female mosquitoes were transferred into mesh-covered paper cups (200 mL, Neptune Paper Cup Co., Newark, NJ, USA), and wet cotton placed on top of the mesh kept mosquitoes hydrated. Cups with mosquitoes were kept in a 16 L cooler box (Coleman Company, Inc. Chicago, IL, USA) until mosquitoes were released into large outdoor screened cages [28] (Figure 4). The cages were 69.25 m^3^ in volume, with dimensions of 3.05 m (front wall height), 2.44 (back wall height) × 6.5 m (length) × 4.2 m (width), and used in trap efficacy trials at USDA-ARS, CMAVE, Gainesville, FL, USA. The assays for attractancy were conducted using two cages, each equipped with a bait station containing a CDC trap (New Standard Miniature Incandescent Light Trap Model 1012, John W. Hock Company, Gainesville, FL, USA), and vortex unit (Vortex-Genie 2 Laboratory Mixer, Daigger Scientific, Inc., Vernon Hills, IL, USA), with no dry-ice, and the lights off (Figure 5). Sheep wool (20 g) was placed in a knee-high stocking (Leggs Everyday Knee Highs, Winston-Salem, NC, USA) and attached to the vortex platform head with a 7.6 cm diameter cover using black steel fold-back binder clips (large, Business Source Products, Post Falls, ID, USA). Vortexer was set at speed level 1 (600 rpm) with continuous hands-free operation. Adult female *Ae. aegypti* mosquitoes (*n* = 200) were used for each experimental cage. Before releasing mosquitoes into the cages, the mosquito traps were turned on and then, depending on the experimental setup, sheep wool in a stocking or only an empty stocking was attached to the vortex head (Figure 5), and the vortex was set on vibration or turned off. Every experiment was conducted in both cages simultaneously at least six times for each pair of baits. The baits were alternated for each subsequent test.

The following setups were used to determine the effects of various combinations of traps baited with/without wool, or with/without vibration. In setup/experiment A, one cage was equipped with wool which was compared to a cage without wool with the Vortex apparatus in both cages turned on to vibrate. In setup/experiment B, both cages were equipped with wool, but the Vortex vibrated in only one cage. For setup/experiment C, only one cage was baited with wool and the Vortex turned on to vibrate, which was compared against a cage with an empty stocking and the Vortex turned off. In setup/experiment D, the Vortex was turned on without wool as bait and compared to a cage containing wool as bait but the Vortex was turned off (no vibration).

Assays were started in the late afternoons at about 5–6 pm and lasted overnight. Next morning, after ~16 h, mosquitoes in the traps were collected, counted, and data analyzed. All experiments in the large outdoor cages were conducted in summer 2018. To compare attractiveness of the paired baits to each other the mean number of mosquitoes mean ± SEM (standard error of the mean) for each bait, and the mean of the difference of these means for each pair ±SED (standard error of difference) were used. Additionally, relative attractiveness (RA) was used to compare attractiveness of the paired baits to each other, similar to the glass tube assays. RA was expressed as mean RA ± SEM, and reported as the number of mosquitoes collected from the trap of one bait, divided by the total number of mosquitoes collected from both (paired) traps together (the maximum attractive value for RA is 1; the efficacies of the paired baits are considered similar if RA = 0.5).

### 2.6. Statistical Analysis

A two-tailed Student’s *t*-test was used to analyze the results for glass tube attractancy assays. A two-tailed paired *t*-test was used to compare the collections of the pared baits in semi-field attractancy bioassays. All statistics were evaluated as significant when there was less than 5% chance of error (*p* < 0.05). Statistical analysis was conducted using RStudio [29] and Excel.

**Table 1 insects-13-00208-t001:** Chemical composition of sheep wool hydrodistillate, and DHS volatile collection analyzed by gas chromatography–mass spectrometry (GC–MS).

Compound		Area %			References
RRI	Name	Hydrodistillate	DHS	Identification	
960	6-methyl-2-heptanone ^b^		1.0	MS	[10,30,31]
993	2-octanone ^a^	0.9	0.7	RRI, MS	[32]
1005	octanal ^a^		1.8	RRI, MS	[8]
1078	1-octanol ^a^		7.0	RRI, MS	[30,31]
1079	*p*-cresol ^a^	1.8		RRI, MS	
1100	linalool		1.7	RRI, MS	[30,33]
1106	nonanal ^a^		15.3	RRI, MS	[8]
1120	3,5-dimethyl-1,2,4-trithiolane (one of the isomers) ^a,b^	2.1		MS	[30,31]
1128	3,5-dimethyl-1,2,4-trithiolane (one of the isomers) ^a,b^	3.7		MS	
1196	5,6-dihydro-2,4,6-trimethyl-4*H*-1,3,5-dithiazine (thialdine, base peak 163) ^a^	86.6		RRI, MS	
1200	dodecane		5.8	RRI, MS	[34]
1207	decanal		1.1	RRI, MS	[30]
1284	bornyl acetate ^b^		1.4	MS	[32,35]
1300	tridecane ^a^		16.2	RRI, MS	[30,36]
1392	longifolene ^a^		10.7	RRI, MS	[34]
1399	tetradecane ^a^		13.6	RRI, MS	[30]
1404	β-caryophyllene ^a^		11.8	RRI, MS	[35]
1498	pentadecane ^a^		5.1	RRI, MS	[30,36]
	Total	95.1	93.2		

^a^ Detected by both methods GC-MS and GC-FID/GC-MS in HS-Hayesep-Q collection, ^b^ Tentative identification from Wiley, DHS, dynamic headspace, RRI: Relative retention indices calculated against *n*-alkanes [22]. Identification method based on the relative retention indices (RRI) of authentic compounds on the DB-5 column; MS, identified based on computer matching of the mass spectra with those of the Wiley and NIST libraries and literature data.

## 3. Results

### 3.1. Extraction of Volatiles and Analysis by GC–MS and GC–FID/GC–MS

Sheep wool volatiles were collected by hydrodistillation with the Clevenger apparatus, and also using HS collection methods, such as dynamic HS extraction by Hayesep-Q polymer adsorbent (Figure 1 and Figure 2), and HS-SPMEs with three different types of fiber coating: PDMS/DVB, CAR/PDMS and PDMS. The chemical compositions of these extracts were identified by GC–MS and GC–FID/GC–MS spectrometry methods (Table 1 and Table 2). Table 1 and Table 2 show RRIs, compound names, composition (%), and literature references [8,10,11,12,30,31,32,33,34,35,36,37,38,39,40,41,42,43] presenting these compounds in connection with the sheep products.

#### 3.1.1. GC–MS and GC–FID/GC–MS Analysis of Hydrodistillate

According to the GC–MS and GC–FID/GC–MS methods, the major component of the hydrodistillate was thialdine (~86%). Presence of thialdine was confirmed by comparing ^1^H and ^13^C NMR shifts of hydrodistillate to neat thialdine. Five compounds were detected in the hydrodistillate by GC–MS and 19 compounds by GC–FID/GC–MS (Table 1 and Table 2). Two isomers of 3,5-dimethyl-1,2,4-trithiolane, and *p*-cresol (0.6–3.7%) were identified by both methods.

#### 3.1.2. GC–MS and GC–FID/GC–MS Analysis of DHS Collection

In the DHS extract with Hayesep-Q adsorbent, 14 compounds were identified by GC-MS (Table 1). Alkanes were the major constituents (~41%): tridecane, tetradecane, dodecane, and pentadecane (listed by descending order); followed by the sesquiterpenes (~22%): longifolene and β-caryophyllene; aldehydes (~18%): nonanal, octanal, and decanal; and one alcohol: 1-octanol (~7%). By GC-FID/GC-MS a total of 18 compounds were identified (Table 2) with alcohols as the major constituents (~41%): 2-ethyl hexanol, 1-octanol, 2-hexanol, ionol, 3-hexanol, 1-heptanol and 1-nonanol; followed by alkanes (~25%): tridecane, pentadecane, tetradecane, hexadecane, octadecane and heptadecane; sesquiterpenes (~13%): longifolene and β-caryophyllene; one aldehyde: nonanal (~11%), and one ketone: 2-hexanone (~8%). A total of 13 compounds in this collection were identified by both GC–MS and GC–FID/GC–MS methods (Table 1 and Table 2).

**Table 2 insects-13-00208-t002:** Chemical composition of sheep wool hydrodistillate, DHS and HS-SPME volatile collections analyzed by gas chromatography–flame ionization detector (FID) combined with GC–MS.

Compound		Area %					Identification	References
RRI	Name	Hydrodistillate	Headspace					
			DHS	HS-SPME				
				PDMS/DVB	CAR/PDMS	PDMS		
1087	2-hexanone ^b^		7.9				MS	[32]
1093	hexanal			9.0	16.9		RRI, MS	[37]
1155	1-butanol			trace	4.8		RRI, MS	[11]
1194	heptanal			13.2	13.9		RRI, MS	[8,37]
1202	3-hexanol ^b^		2.7				MS	[37]
1212	isoamyl alcohol (=3-methyl-1-butanol) ^b^			5.2	12.1		MS	-
1222	2-hexanol ^b^		4.6				MS	[36]
1260	1-pentanol			3.0	6.4		RRI, MS	[32]
1290	2-octanone ^a,b^			1.6	1.0		MS	
1296	octanal ^a^			5.0	3.2		RRI, MS	[8,37]
1300	tridecane ^a^		10.3				RRI, MS	[30]
1360	1-hexanol			6.9	9.6		RRI, MS	[32]
1400	tetradecane ^a^		3.1				RRI, MS	[36]
1400	nonanal ^a^		10.6	8.0	1.9	6.6	RRI, MS	[8,37]
1412	(*E*)-2-hexenol ^b^			1.3	1.0		MS	-
1463	1-heptanol		2.4	3.3	3.4		RRI, MS	[35]
1496	2-ethyl hexanol ^a,b^		21.0	3.9	3.2		MS	[10]
1496	2-decanone ^b^	0.5					MS	[32]
1500	pentadecane ^a^	0.5	5.4				RRI, MS	[30,36]
1541	benzaldehyde			1.0	0.5		RRI, MS	[11]
1562	1-octanol ^a^	0.4	5.7	2.2	0.9		RRI, MS	[36]
1583	longifolene (=junipene) ^a,b^		6.9				MS	[34]
1600	hexadecane	0.3	3.0				RRI, MS	[30,36]
1604	2-undecanone ^b^	0.2					MS	[32]
1612	β-caryophyllene ^a^		6.2				RRI, MS	[35]
1614	3,5-dimethyl-1,2,4-trithiolane (one of the isomers) ^a,b^	0.6					MS	[30,31]
1631	γ-pentalactone (=γ-valerolactone) ^b^			1.1	0.7	-	MS	[39]
1634	3,5-dimethyl-1,2,4-trithiolane (one of the isomers) ^a,b^	0.6					MS	[30,31]
1664	1-nonanol	0.9	1.3	0.5	-	-	RRI, MS	[12]
1674	2-methylbutanoic acid ^b^			2.1	3.1	3.5	MS	[37]
1700	heptadecane		1.1				RRI, MS	[30,36]
1706	α-terpineol			0.6	-	-	RRI, MS	-
1762	5,6-dihydro-2,4,6-trimethyl-4*H*-1,3,5-dithiazine (thialdine, base peak 163) ^a^	85.1					RRI, MS	[30,33]
1800	octadecane		2.4				RRI, MS	[30,36]
1896	benzyl alcohol			0.3	-	-	RRI, MS	-
1900	nonadecane		0.5				RRI, MS	[36]
1815	2-tridecanone ^b^	0.1					MS	[37]
1925	ionol		3.7	9.7	6.6	81.1	RRI, MS	-
1973	1-dodecanol	0.1					RRI, MS	[39]
2033	phenol			0.7	0.5	-	RRI, MS	[40]
2077	1-tridecanol	0.1					RRI, MS	[41]
2094	*p*-cresol ^a^	1.1		12.8	5.7	8.8	RRI, MS	[30,31]
2102	*m*-cresol ^b^			1.0	0.2	-	MS	[42]
2131	hexahydrofarnesyl acetone ^b^	2.0					MS	-
2132	5,6-dihydro-2,4-dipentyl-4*H*-1,3,5-dithiazine ^b^ (base peak 126)	0.6					MS	-
2179	1-tetradecanol	0.1					RRI, MS	[12]
2279	1-pentadecanol	2.5					RRI, MS	
2384	1-hexadecanol	2.4					RRI, MS	
2475	1-heptadecanol	0.1					RRI, MS	[42]
	Total	98.2	98.8	92.4	95.6	100.0		

^a^ Detected by both methods GC–MS and GC–FID in HS-Hayesep-Q collection, ^b^ Tentative identification from Wiley, RRI: Relative retention indices calculated against *n*-alkanes, % calculated from FID data; Identification method based on the relative retention indices (RRI) of authentic compounds on the HP Innowax column; MS, identified based on computer matching of the mass spectra with those of the Wiley and MassFinder libraries and comparison with literature data, PDMS/DVB: precoated 65 μm-thick layer of polydimethylsiloxane-divinylbenzene, CAR/PDMS: precoated 75 μm-thick layer of carboxen-polydimethylsiloxane Red fiber: precoated PDMS: precoated 100 μm thick layer of polydimethylsiloxane.

#### 3.1.3. GC–FID/GC–MS Analysis of HS-SPME Extracts by Three Different Fibers

In the HS-SPME collection with PDMS/DVB fibers, 23 compounds were detected (Table 2). The extract consisted mostly of alcohols (~49%): *p*-cresol, ionol, 1-hexanol, isoamyl alcohol, 2-ethyl hexanol, 1-heptanol, 1-pentanol, 1-octanol, and *m*-cresol; aldehydes (~36%): heptanal, hexanal, nonanal, octanal, and benzaldehyde; also, 2-octanone (1.6%), γ-pentalactone (1.1%), and 2-methylbutanoic acid (2.1%). Almost the same major ingredients, 20 compounds, were detected in the extract by CAR/PDMS fibers; alcohols (~56%): isoamyl alcohol, 1-hexanol, ionol, 1-pentanol, *p*-cresol, 1-butanol, 1-heptanol, 2-ethyl hexanol, and (*E*)-2-hexenol; aldehydes (~36%): hexanal, heptanal, octanal, and nonanal; also, 2-octanone (1.0%), and 2-methylbutanoic acid (3.1%). Only four compounds were detected in the collection by PDMS fibers, with the major constituent ionol (81%), followed by *p*-cresol (8.8%), nonanal (6.6%) and 2-methylbutanoic acid (3.5%); All four of these compounds were detected in the extracts collected by PDMS/DVB and CAR/PDMS absorbents, as well.

### 3.2. Bioassays and Data Analysis

#### 3.2.1. Attractant Activity of Sheep Wool Hydrodistillate and Thialdine in Glass Tubes

Glass tube assays were conducted using sheep wool hydrodistillate in comparison with its major component, thialdine (Figure 3, Table 3). A slight attractant effect was observed at 0.1 µg/cm^2^ of sheep wool hydrodistillate (60 min) or thialdine (15 min); little or no attractancy was observed on sheep wool hydrodistillate or thialdine at all other tested concentrations and times (Table 3), especially when compared to 1 µg/cm^2^ of 1-octen-3-ol (0.75 ± 0.02 at 15 min; 0.75 ± 0.02 at 30 min; 0.81 ± 0.05 at 60 min). Similarly, co-application with 10 mL of CO_2_ showed little attractancy at any time point to *Ae. aegypti* mosquitoes when tested at 0.1 and 0.01 µg/cm^2^ of sheep wool hydrodistillate. Thus, according to the assay results, the hydrodistillate and thialdine did not show any significant attractancy against female adult *Ae. aegypti*. It should be noted, that a slight repellent effect was observed at 10 µg/cm^2^ of sheep wool hydrodistillate at 15 min; and at 1 µg/cm^2^ of thialdine at 30 min.

#### 3.2.2. The Attractant Activity of Sheep Wool Using Semi-Field Bioassays

A two-tailed paired *t*-test was used to compare the efficacies of the baits and determine the statistical significance of the differences within the pairs. The mean (±SEM) number of mosquitoes for each bait and the mean of the difference of these means (±SED) for each pair are presented in Table 4. The difference was statistically very significant (60.29 ± 15.30) when bait “vibration + wool” was compared to bait “none” (setup C) (Figure 4 and Figure 5). The differences between the paired baits “vibration + wool” vs. “vibration” 35.25 ± 8.85 (setup A), and “vibration + wool” vs. “wool” 29.67 ± 7.62 (setup B) were significant, as well. There was no significant difference between the efficacies of captures by baits “vibration” vs. “wool” 20.83 ± 20.82 (D), Table 4.

Relative attractiveness for paired baits “vibration + wool” vs. “none” was 0.70 ± 0.04 (setup C). RAs for paired baits “vibration + wool” vs. “vibration”, and “vibration + wool” vs. “wool” were almost identical, 0.58 ± 0.02, and 0.58 ± 0.03, correspondingly (setups A, and B).

These results show that vibrated sheep wool is a better attractant than the baits with only wool or vibrated bait but without wool. Baits “vibration” vs. “wool” seem to have almost similar effects.

## 4. Discussion

### 4.1. Comparison of the Extraction Techniques for Chemical Analysis

The chemical composition of the sheep wool extracts varied across the collections depending on the extraction methods or types of fibers used: e.g., the hydrodistillate mainly contained thialdine, while it was not detected in HS-volatile collections. Thialdine previously was reported in the volatiles of roasted lamb fat [30,33], and preserved duck eggs [44]. Thialdine is formed during cooking and is most familiar as a flavor component of foods [45]. Investigation using X-ray diffraction data showed that thialdine has an all *cis*-configuration [46].

3,5-Dimethyl-1,2,4-trithiolane isomers were also characteristic of hydrodistillate and detected by both MS methods. Hexahydrofarnesyl acetone, and alcohols: 1- do-, tri-, tetra-, penta-, hexa-, and hepta-decanols, with a few other trace compounds were detected in hydrodistillate by GC–FID/GC–MS (Table 1 and Table 2). GC–FID/GC–MS also found *p*-Cresol, hexadecane, pentadecane, 1-octanol, and 1-nonanol in hydrodistillate, and in at least one of the HS collections.

For DHS extract, almost all compounds identified by GC–MS were detected by GC–FID/GC–MS, as well, except for linalool, dodecane, and decanal; while, with GC–FID/GC–MS, additionally, 2-hexanone, 3-hexanol, 2-hexanol, 1-heptanol, hexadecane, 1-nonanol, heptadecane, octadecane, nonadecane and ionol were detected (Table 2).

Nonanal was the only compound found in all HS collections, detected by both MS methods. 1-Heptanol, 2-ethyl hexanol, and 1-octanol were also found in all HS collections except for SPME by PDMS. The volatiles collected using HS-SPME by PDMS/DVB, and CAR/PDMS, such as hexanal, 1-butanol, heptanal, isoamyl alcohol, 1-pentanol, 1-hexanol, (*E*)-2-hexenol, benzaldehyde, γ-pentalactone, 2-methylbutanoic acid, phenol, *m*-cresol, were not detected in DHS collection.

While the major component of wool hydrodistillate was thialdine, the major components for HS-SPME (by PDMS/DVB, CAR/PDMS, and PDMS fibers), and DHS (by Hayesep-Q fiber) collections of row wool were alcohols, aldehydes, alkanes and sesquiterpenes. The HS extracts did not contain thialdine, since this compound is formed during cooking (in this case, hydrodistillation) [45].

It is known that the composition of extracts could greatly depend on the extraction methods used, e.g., in HS extracts the volatile collections vary according to the type of fiber coating, and its ability to absorb/adsorb volatiles; also, desorption techniques (e.g., with solvent, thermal) can play an important role on the composition of the extract. HS SPME and DHS are convenient methods for determination of volatile compounds, and both require much smaller sample amounts compared to hydrodistillation; also, HS SPME and DHS methods require no solvents to absorb/adsorb volatiles, and this is an important advantage. Hydrodistillation can extract large amounts of semi-volatile constituents, most of them might not be in sufficient amounts for headspace collection; hydrodistillation uses water steam, and sometimes this can induce chemical reactions (e.g., thialdine formation).

Furthermore, when concentration of the extracted volatiles is very low, the spectrometer cannot detect compounds at concentrations that are below its detection levels. The wool material we used in this study was stored and, accordingly, the volatiles emission might not be as intensive as for fresh material. Accordingly, the volatiles composition varied greatly not only in terms of the extraction techniques used, but also by detection methods applied. Although the two methods, GC–MS and GC–FID/GC–MS, have advantages and disadvantages [47], both analytical approaches provided important insights. Thus, according to the analytical results (Table 1 and Table 2), none of the methods were ideal for extraction, or detection of the volatiles, rather they complemented each other.

A total of 52 compounds were detected (Table 1 and Table 2), of which seven compounds: isoamyl alcohol, (*E*)-2-hexenol, α-terpineol, benzyl alcohol, ionol, hexahydrofarnesyl acetone, and 5,6-dihydro-2,4-dipentyl-4*H*-1,3,5-dithiazine were not reported previously in sheep products to the best of our knowledge.

### 4.2. Extracted Sheep Wool Volatiles and Their Activity against Mosquitoes

Most of the volatile compounds we identified from sheep wool (Table 1 and Table 2) were found previously in sheep-derived products and reported in the literature [8,10,11,12,30,31,32,33,34,35,36,37,38,39,40,41,42,43]. This set includes those aldehydes, potent attractants for RVFV mosquito vectors—heptanal, octanal, nonanal, decanal [8]. Among the semiochemicals used by mosquitoes during mating, oviposition, host-seeking, and/or sugar feeding are the compounds: 2-ethyl hexanol, *p*-cresol, hexanal, heptanal, octanal, nonanal, decanal, benzaldehyde, β-caryophyllene, and phenol [8,48,49,50,51,52,53,54,55,56,57,58,59]. Various insects including mosquitoes were attracted to dodecane [59]. 2-Hexanone and 1-butanol showed mosquito attractant activity [60], and 2-octanone showed attractancy against *Culex pipiens* (Diptera: Culicidae) [61]. Octanal was attractive to *Ae. aegypti* mosquitoes [62] and octanal, nonanal, and decanal, in olfactometer trials, interfered with the attraction of species to a host [63]. Torres-Estrada et al. (2005) identified several compounds from plant extracts, including longifolene and caryophyllene, as attractants for oviposition of *Anopeles albimanus* (Diptera: Culicidae) [64]. An increased attraction of *An. stephensi* (Diptera: Culicidae) was observed for 3-methylbutanoic acid, 2-methylbutanoic acid, hexanoic acid, and tridecane [65]. Hexadecane along with a group of other compounds significantly reduced trap catches of *An. arabiensis* (Diptera: Culicidae) compared to a negative control [66].

β-Caryophyllene and decanal elicited antennal responses from *An. arabiensis* during gas chromatography coupled to electroantennographic detection (GC–EAD) experiments [48]. Electroantennograms recorded with *Toxorhynchites moctezuma* (Diptera: Culicidae) and *T. amboinensis* (Diptera: Culicidae) mosquitoes for phenol, *m*-cresol, *p*-cresol (along with several other compounds) showed oviposition attractancy [67]. Hexanal showed EAD-activity against *An. gambiae* (Diptera: Culicidae) [51].

The study conducted by Robinson et al. [53] revealed an increase in attraction of *An. gambiae* towards a synthetic blend containing heptanal, nonanal and, octanal. The compounds 3-hexanol, 1-hexanol, and 1-pentanol activate CO_2_-sensitive olfactory neurons in *Ae. aegypti* and *An. gambiae* [68]. Higher amounts of 2-methylbutanoic acid and octanal, along with two other compounds, were associated with individuals that were highly attractive to *An. gambiae* [69]. Isoamyl alcohol was among the six most promising candidate oviposition semiochemicals (*An. gambiae*) out of 50 identified volatiles that were emitted from the headspace of the most attractive bacteria isolates [70].

According to these literature studies [8,10,11,12,30,31,32,33,34,35,36,37,38,39,40,41,42,43,48,49,50,51,52,53,54,55,56,57,58,59,60,61,62,63,64,65,66,67,68,69,70], many constituents of sheep wool (Table 1 and Table 2), showed attractant activity against mosquitoes. Aldehydes that were active against mosquitoes: hexanal, heptanal, octanal, nonanal, decanal, and benzaldehyde [8,10,11,12,30,31,32,33,34,35,36,37,38,39,40,41,42,43,48,49,50,51,52,53,54,55,56,57,58,62,63,69] were present in a DHS extract of sheep wool at ~18.2% (nonanal, octanal, and decanal), and ~10.6% (nonanal), detected by GC–MS, and GC–FID/GC–MS, correspondingly; in HS-SPME collections, by PDMS/DVB, and CAR/PDMS they were present at ~36.2% and 36.4% (heptanal, hexanal, nonanal, octanal, and benzaldehyde), correspondingly, and by PDMS ~ 6.6% (nonanal), detected by GC–FID/GC-MS (Table 1 and Table 2).

Alcohols that were active against mosquitoes: 1-butanol, *m*-cresol, *p*-cresol, 1-hexanol, 3-hexanol, 2-ethyl hexanol, isoamyl alcohol, phenol, 1-pentanol [60,67,68,70], were present in hydrodistillate at ~1.8% and ~1.1% (*p*-cresol), detected by GC–MS, and GC–FID/GC–MS, correspondingly; in DHS extract they were at ~23.7% (2-ethyl hexanol, and 3-hexanol), and, in HS-SPME collections, by PDMS/DVB, CAR/PDMS, and PDMS at ~32.8% (*p*-cresol, 1-hexanol, isoamyl alcohol, 2-ethyl hexanol, 1-pentanol, and *m*-cresol), ~42.0% (isoamyl alcohol, 1-hexanol, 1-pentanol, *p*-cresol, 1-butanol, 2-ethyl hexanol, and *m*-cresol), and ~8.8% (*p*-cresol), correspondingly, detected by GC–FID/GC–MS (Table 1 and Table 2).

Sesquiterpenes that were active against mosquitoes: β-caryophyllene, longifolene, [48,70], were present in DHS extract at ~22.5%, and ~13.1% (β-caryophyllene, longifolene) detected by GC–MS, and GC–FID/GC–MS, correspondingly (Table 1 and Table 2).

Alkanes that were active against mosquitoes: dodecane, tridecane, hexadecane [59,65,66], were present in DHS extract at ~22.0% (tridecane, and dodecane), and ~13.3% (tridecane, and hexadecane) detected by GC–MS, and GC–FID/GC–MS, correspondingly; in hydrodistillate hexadecane was detected at ~0.3% by GC–FID/GC–MS (Table 1 and Table 2).

Ketones that were active against mosquitoes: 2-hexanone, and 2-octanone [60,61], were present in hydrodistillate and DHS extract at ~0.9%, and 0.7% (2-octanone), correspondingly, detected by GC–MS; and detected by GC–FID/GC–MS, in DHS extract they were present at ~7.9% (2-hexanone), and, in HS-SPME collections, by PDMS/DVB and CAR/PDMS at ~1.6%, and ~1.0% (2-octanone), correspondingly, (Table 1 and Table 2).

2-Methylbutanoic acid which showed activity against mosquitoes [65,69], was present in HS-SPME collections by PDMS/DVB, CAR/PDMS, and PDMS at ~2.1%, 3.1%, and 3.5%, correspondingly, detected by GC–FID/GC–MS (Table 2).

### 4.3. Bioassays

While the sheep wool hydrodistillate and its major component thialdine did not show any significant attractant activity against adult female *Ae. aegypti* mosquitoes in glass tube bioassays, the baits with dynamic (vibrated) raw sheep wool were attractive to *Ae. aegypti* compared to the static baits, or dynamic baits but without wool. Baits with sheep wool, dynamic or static, could both be sources of volatiles, but under static conditions the odor emanating from wool may not be strong enough to stimulate the mosquito’s olfactory sensors. When vibrated, the volatile compounds, most probably, are released more intensively from the wool fiber, and the released volatiles, as potential attractants, increase mosquito capture. The attractant activity of vibrated raw sheep wool could be attributed to the volatile molecules from the HS extracts containing significant amounts of those aldehydes, alcohols, sesquiterpenes, alkanes, ketones, or 2-methylbutanoic acid (Table 1 and Table 2) that were identified as active against mosquitoes according to the literature reports [8,10,11,12,30,31,32,33,34,35,36,37,38,39,40,41,42,43,48,49,50,51,52,53,54,55,56,57,58,59,60,61,62,63,64,65,66,67,68,69,70]. On the contrary, in sheep wool hydrodistillate these compounds were present only in small amounts: e.g., *p*-cresol, detected by GC–MS and GC–FID/GC–MS, was present at <~2%, 2-octanone, detected by GC–MS, was present at ~0.9%, and hexadecane, detected by GC–FID/GC–MS, was present at ~0.3%. The major component of the hydrodistillate, thialdine, which is product of cooking [45], was not attractive to adult female *Ae. aegypti,* according to glass tube bioassays.

In semi-field bioassays with raw sheep wool, the following factors could be involved affecting mosquito attraction: chemical (volatiles released by wool) and visual (vibrating Vortex) signals. It should be noted that sound could also influence mosquito behavior, according to Dou et al. [71], who studied *Ae. aegypti* and *An. gambiae* mosquitoes responses to sound stimulus; in our case, sound from the vibrating Vortex.

Thus, the attraction of adult female *Ae. aegypti* to vibrating sheep wool may have been a combination of the effects of olfactory, visual, and, also, acoustic cues; of which the olfactory cues should be most important as, according to a recent study on adult female *Ae. aegypti* by Vinauger et al. [72], olfactory cues seem to play a greater role in attracting mosquitoes to the host than their visual cues.

According to Tchouassi et al. [7], during storage the hair could lose volatile compounds over time compared to the dynamic production from live animals, and in our semi-field experiments, we tried to create the dynamic conditions by vibrating sheep wool, which, we assume, intensified the process of releasing volatiles, that appear to be still present in stored sheep wool, and this intensive emission of volatiles considerably increased mosquito collection in the traps.

## 5. Conclusions

In this work, we studied the chemical composition of stored sheep wool volatiles and the attractancy of wool for adult female *Ae. aegypti* mosquitoes. The wool extracts were obtained by hydrodistillation and the various HS extraction methods. A total of 52 compounds were still present in stored wool detected by GC–MS and GC–FID/GC–MS methods. Seven compounds were not reported previously in the sheep products to the best of our knowledge. These compounds were collected by various extraction methods, and were detected with one and/or another spectrometry techniques, so the methods complemented each other. The hydrodistillate mainly contained thialdine detected by both spectrometry techniques. The main components of DHS volatile extract identified by GC–MS were alkanes, and, by GC–FID/GC–MS were alcohols. The main components of HS-SPME extracts identified by GC–FID/GC–MS in PDMS/DVB, and C/PDMS collections were alcohols and aldehydes, while ionol was the main component when PDMS was used as absorbent. Many of the detected compounds were previously reported as having some level of attractancy against mosquitoes. The sheep wool hydrodistillate and its main component thialdine did not show any significant attractant activity in the glass tube tests. Semi-field bioassays in the outdoor large screened cages equipped with the CDC traps and Vortex apparatus revealed the bait with vibrated sheep wool was a significantly better attractant (RA ~0.70) than the bait with no wool and no vibration. In baits with vibrated sheep wool, attraction to mosquitoes could be influenced by combination of olfactory, and visual cues (also, sound from vibrated Vortex might influence mosquito behavior). Of these, olfactory signals seem to be the most important. We assume that under dynamic conditions stored sheep wool more intensively releases volatiles while the emission is limited when wool is not moving and this approach could be an option for increasing the efficacy of mosquito capture in bait systems. In the future, on the basis of identified potential attractants from the raw sheep wool volatiles, various blends could be tested for attractancy against mosquitoes, in order to find new active volatile combinations that increase mosquito capture in the traps. Research on raw sheep wool could be extended to test sheep wool baits with CO_2_, and light; and experiments could be conducted with other natural materials. Sheep wool is an easily available, affordable, and environmentally friendly product that should make this material attractive for use in mosquito management and surveillance.

## Figures and Tables

**Figure 1 insects-13-00208-f001:**
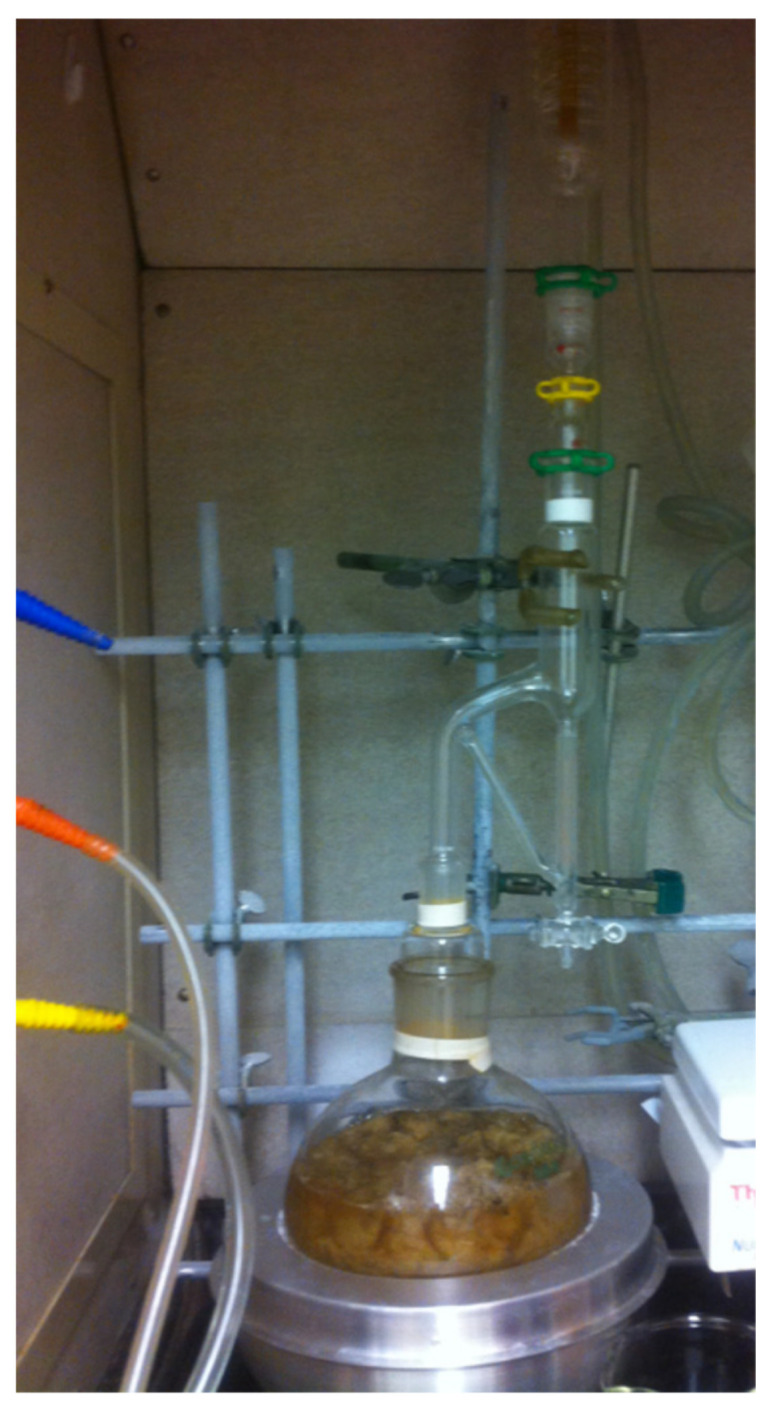
Hydrodistillation of sheep wool using the Clevenger type setup.

**Figure 2 insects-13-00208-f002:**
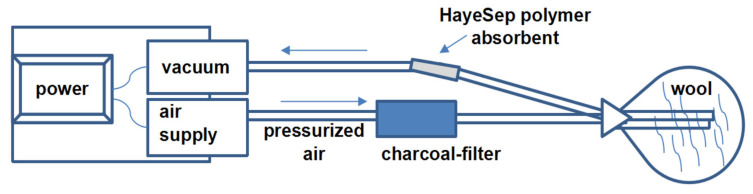
A schematic diagram of the dynamic headspace collection (DHS) system for collection of wool volatiles using Hayesep-Q polymer adsorbent.

**Figure 3 insects-13-00208-f003:**
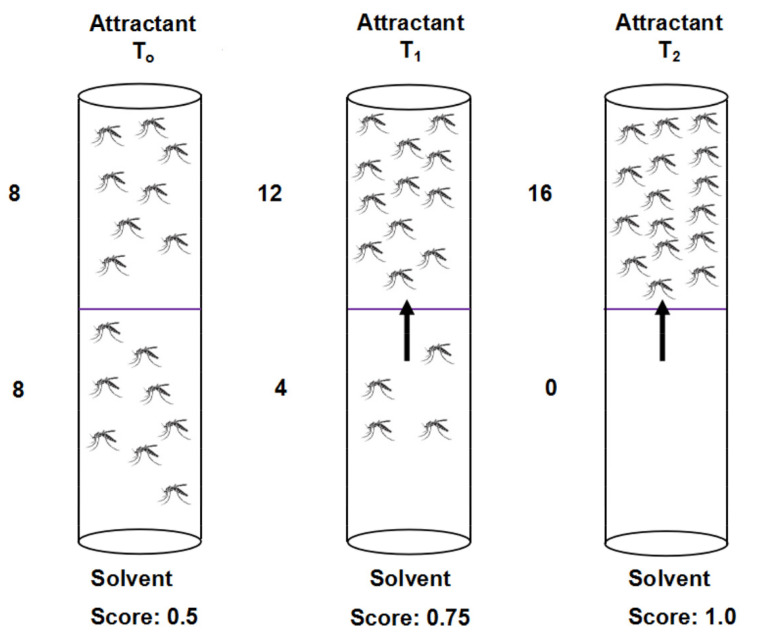
Attractancy assay for mosquitoes in glass tubes containing 16 females. Solvent or sheep hair volatiles were applied to filter papers and movement of mosquitoes was monitored over 1 h. Scoring of attractancy was as indicated. T_o_, T_1_, and T_2_, treatments.

**Figure 4 insects-13-00208-f004:**
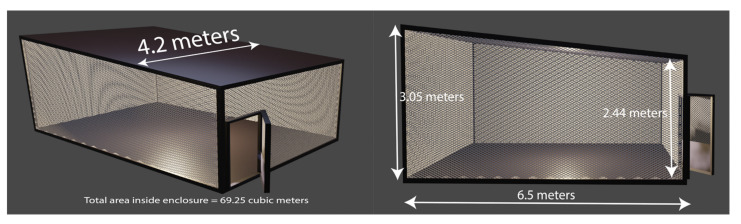
A diagram for the large screened outdoor cage. The bioassay setup is placed in the center of the cage.

**Figure 5 insects-13-00208-f005:**
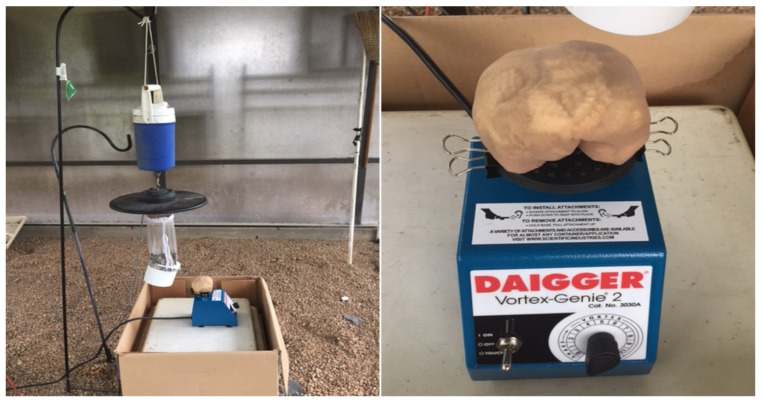
CDC trap and sheep wool placed in women’s stocking attached to the Vortex platform head.

**Table 3 insects-13-00208-t003:** Proportion (±SE) of adult female *Ae. aegypti* mosquitoes to sheep wool hydrodistillate and thialdine in glass tube assay.

Concentrations	Sheep Wool Hydrodistillate			Thialdine		
(µg/cm^2^)	15 min	30 min	60 min	15 min	30 min	60 min
10	0.38 ± 0.06	0.48 ± 0.04	0.42 ± 0.02	0.46 ± 0.04	0.50 ± 0.03	0.48 ± 0.02
1	0.44 ± 0.06	0.48 ± 0.02	0.48 ± 0.02	0.54 ± 0.08	0.38 ± 0.04	0.44 ± 0.03
0.1	0.58 ± 0.04	0.58 ± 0.04	0.69 ± 0.06	0.62 ± 0.06	0.52 ± 0.02	0.58 ± 0.02
0.01	0.58 ± 0.04	0.58 ± 0.04	0.48 ± 0.05	0.50 ± 0.03	0.50 ± 0.06	0.50 ± 0.04
Solvent control	0.52 ± 0.02	0.54 ± 0.02	0.50 ± 0.03	0.52 ± 0.02	0.54 ± 0.02	0.48 ± 0.02

SE, standard error.

**Table 4 insects-13-00208-t004:** Summary of paired *t*-tests for comparison of differences between the mean amounts of mosquitoes collected in the paired traps in setups A, B, C, and D in semi-field assay.

Setups	Paired Bates	Mean ± SEM	Mean of Difference ± SED	95% CI of Difference	t-Value (*t*-Critical)	*n*, df	*p*-Value Two-Tail
A	vibration + wool	126.13 ± 11.91	35.25 ± 8.85	14.31–56.19	3.98 (2.36)	8, 7	0.005
	vibration	90.88 ± 11.13					
B	vibration + wool	113.5 ± 12.82	29.67 ± 7.62	10.08–49.25	3.90 (2.57)	6, 5	0.012
	wool	83.83 ± 15.39					
C	vibration + wool	98.29 ± 19.56	60.29 ± 15.30	22.86–97.72	3.94 (2.45)	7, 6	0.008
	none	38.00 ± 6.70					
D	vibration	151.67 ± 13.05	20.83 ± 20.82	(−2.09)–43.75	2.34 (2.57)	6, 5	0.067
	wool	130.83 ± 13.87					

SEM, standard error of the mean, SED, standard error of difference, CI, confidence interval, *n*, number of paired treatments.

## Data Availability

The data presented in this study are available on request from the corresponding author.

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
