# Peer review of "Studies on the Volatiles Composition of Stored Sheep Wool, and Attractancy toward Aedes aegypti Mosquitoes"

_insects, 2022, doi:10.3390/insects13020208_

Round 1

Reviewer 1 Report

  1. Title shall be changed - present title indicates two different statements.
  2. Keywords may be reduced to 5/6 words removing those already present in the title.
  3. Introduction - a paragraph indicating the economical and health impacts of Ae. aegypti shall be added along with scope of work. Further a clear picture of mosquito resistance and advantage of eco-friendly control, through natural substance should be provided to justify your work. At least refer following documents and include (10.3390/molecules26123695, 10.1016/j.bcab.2021.101915 )
  4. The reason to employ sheep wool shall be added in intro - the feasibility and also the reliability of sheep wool as a source shall be justified.

5.Procedures shall be substantiated with appropriate citations.

  1. Certain results are not discussed. presenting results and discussion as separate sections would shed more light on the inferences of the work.

Author Response

Response to Reviewer 1 Comments

Point 1: Title shall be changed - present title indicates two different statements.

Response 1: We thank the reviewer for this comment. We have changed the title “Stored Sheep Wool Volatiles and Efficacy of Mosquito Capture” with “Studies on the Stored Sheep Wool Volatiles Composition, and Attractancy Against Aedes aegypti Mosquitoes”

Point 2: Keywords may be reduced to 5/6 words removing those already present in the title.

Response 2: We have removed the keywords: “Aedes aegypti “, “volatiles”, “wool”, and “thialdine”.

Point 3: Introduction - a paragraph indicating the economical and health impacts of Ae. aegypti shall be added along with scope of work. Further a clear picture of mosquito resistance and advantage of eco-friendly control, through natural substance should be provided to justify your work. At least refer following documents and include (10.3389/fphys.2019.01591, 10.1016/j.envint.2017.12.038, 10.3390/molecules26123695, 10.1016/j.bcab.2021.101915 )

Response 3: Introduction, first paragraph: We have added a paragraph considering the comment and adding the suggested reference along with other references.

Point 4: The reason to employ sheep wool shall be added in intro - the feasibility and also the reliability of sheep wool as a source shall be justified.

Response 4: We thank the reviewer for this comment. 

  • Second paragraph from the endof Introduction: We have added e sentence with a citation [14]:

“Sheep wool is an agricultural waste produced by sheep breeding; it is renewable, recyclable and environmentally friendly material, used in many fields and sectors, offering significant benefits for sustainability [14].”

  • At the end of introduction:We have added:

“The molecular composition of sheep wool volatiles, and, correspondingly, the potential activity against insects, could vary depending on the breed, species, age, diet, location, and etc. Evaluation of feasibility and reliability of this material, as source of the desired volatiles should be assessed taking into account all the variabilities, and is out of scope of this work.”

Point 5: Procedures shall be substantiated with appropriate citations.

Response 5: Thank you for pointing out the missing reference. We have added the citations to:.

  • In 2.1. Hydrodistillation. Line 135:

“...system to cool [16,17].”

  • In 1.2. Dynamic headspace collection (DHS). Line 145:

 “…packed in a glass volatiles collection tube [18].”

  • In 2.3. Headspace (HS) collection… Line 169-171:

“... used for extraction of the sheep wool volatiles [19,20]. Sheep wool (0.2 mg) was transferred to a 15 mL vial and sealed with parafilm [21].” 

  • In 2.5.2. Semi-field attractancy assays…” Line 280:

“...large outdoor screened cages [28] (Figure 4).”

Point 6: Certain results are not discussed. presenting results and discussion as separate sections would shed more light on the inferences of the work.

Response 6: We thank the reviewer for this comment.

  • We have separated Results and Discussion.
  • We have moved to Discussion sections

- “3.1.4. (now 4.1) Comparison of the extraction techniques….”,

- “3.1.5. (now 4.2) Extracted sheep wool volatiles and their …”,

- And last paragraph of “3.2.2. (now in section: 4.3 Bioassays) The attractant activity of sheep…”  

  • We have added several paragraphs to Discussion. Lines 514-566:

In section 4.2: last several paragraphs added (highlighted)

In section 4.3: 1.5 paragraph added.

Reviewer 2 Report

The authors investigated, both in laboratory and in semi-field tests, commercially available stored sheep wool for attractancy against Aedes aegypti female mosquitoes. The volatiles from wool were extracted by various techniques. The extracts were analyzed by gas chromatography.They detected fifty two volatile compounds, some of them are already used commercially as mosquito repellents (octanal, linalool etc.) Of those 52 volatile chemicals, seven compounds were not previously reported in sheep wool products ( “to the best of authors knowledge”).

The authors bioasssayed (using glass tubes technique) the sheep wool hydrodistillate and its main component thiadine but that component did not show any significant attractant activity against used Ae.aegypti female mosquitoes.On the other hand, in semi-field bioassays performed in large screened cages (69m3), samples of sheep wool covered by women´s stocking and vibrated by the Vortex platform head (for forced release of volatile compounds) showed higher attractant activity compared with the bait without wool (in an empty stoking) or not vibrated.

Sheep wool is promising, easily available, inexpensive (than some other commercially available attractants) and environmentally friendly material which may be used as an attractant in traps for mosquito monitoring and surveillance and/or in mosquito control. Sheep wool might be more simple material used for attractant in trapping methods compared with e.g. with dry ice (easier for the logistics). Sheep wool (or its components) might be used for long time (days) mosquito collections if they are needed.

The survey of literature is appropriate, the self citations or “honorary citations” are not included.

The submitted article complies with orientation of the MDPI Insects and follows its standards.

I recommend the submitted manuscript for publication in the MDPI Journal.

Author Response

Response to Reviewer 2 Comments

Point 1: The authors investigated, both in laboratory and in semi-field tests, commercially available stored sheep wool for attractancy against Aedes aegypti female mosquitoes. The volatiles from wool were extracted by various techniques. The extracts were analyzed by gas chromatography.They detected fifty two volatile compounds, some of them are already used commercially as mosquito repellents (octanal, linalool etc.) Of those 52 volatile chemicals, seven compounds were not previously reported in sheep wool products ( “to the best of authors knowledge”).

The authors bioasssayed (using glass tubes technique) the sheep wool hydrodistillate and its main component thiadine but that component did not show any significant attractant activity against used Ae.aegypti female mosquitoes.On the other hand, in semi-field bioassays performed in large screened cages (69m3), samples of sheep wool covered by women´s stocking and vibrated by the Vortex platform head (for forced release of volatile compounds) showed higher attractant activity compared with the bait without wool (in an empty stoking) or not vibrated.

 Sheep wool is promising, easily available, inexpensive (than some other commercially available attractants) and environmentally friendly material which may be used as an attractant in traps for mosquito monitoring and surveillance and/or in mosquito control. Sheep wool might be more simple material used for attractant in trapping methods compared with e.g. with dry ice (easier for the logistics). Sheep wool (or its components) might be used for long time (days) mosquito collections if they are needed.

The survey of literature is appropriate, the self citations or “honorary citations” are not included. 

The submitted article complies with orientation of the MDPI Insects and follows its standards.

recommend the submitted manuscript for publication in the MDPI Journal.

Response 1: The authors would like to thank the reviewer for the comment.

Reviewer 3 Report

The authors investigate the volatile compounds of stored sheep wool and explore the attractive effect on mosquitoes. The design of experiments are well described and illustrated. I’m not able to judge the chemical part but it looks exhaustive. The entomological part is clear and easy to follow by the readers. Even if a significant attractive effect has been shown only in the semi-field experiment, I believe the whole results are worth to be published.

My only comment is regarding the semi-field experiment. A miniature CDC light trap has been used and according the picture in figure 3, the trap has been hold on to an insulated container usually used for CO2-released from dry ice. As volatile compounds are known to have a synergetic effect with CO2 regarding attractive effect, did you use dry-ice looking for such cumulative effect? Similarly, as the trap model used include an incandescent light, did the light was on or off during the experiment? It’s not mentioned in your protocol, but according the trap model used and the picture shown, I believe both the use of dry-ice and the light status are worth to be mentioned in material and methods section.

Here is a minor comment: Lines 60-64, the reference [4] should be moved right after the citation of Vasta et al.

Author Response

Response to Reviewer 3 Comments

Point 1: The authors investigate the volatile compounds of stored sheep wool and explore the attractive effect on mosquitoes. The design of experiments are well described and illustrated. I’m not able to judge the chemical part but it looks exhaustive. The entomological part is clear and easy to follow by the readers. Even if a significant attractive effect has been shown only in the semi-field experiment, I believe the whole results are worth to be published.

Response 1: The authors would like to thank the reviewer for the comment.

Point 2: My only comment is regarding the semi-field experiment. A miniature CDC light trap has been used and according the picture in figure 3, the trap has been hold on to an insulated container usually used for CO2-released from dry ice. As volatile compounds are known to have a synergetic effect with CO2 regarding attractive effect, did you use dry-ice looking for such cumulative effect? Similarly, as the trap model used include an incandescent light, did the light was on or off during the experiment? It’s not mentioned in your protocol, but according the trap model used and the picture shown, I believe both the use of dry-ice and the light status are worth to be mentioned in material and methods section.

Response 2:  We have added in the procedure in section 2.5.2. Line 288

“...with no dry-ice, and the lights off…” 

Point 3: Here is a minor comment: Lines 60-64, the reference [4] should be moved right after the citation of Vasta et al.

Response 3: In introduction. Line 84 Reference 4 is reference 10 now)

“... collected volatiles can be affected by various factors; e.g., Vasta et al., [10] using…”

Reviewer 4 Report

The manuscript deals with an interesting topic. However, overall it is pretty cumbersome, unclear in methodology and with poor results as also admit the same authors (see lines 365-367).

These aspects make the paper difficult to read and hard to understand.

In introduction before the authors claim about their research, they should contextualize it much more in depth. Why they decided to carry out this experiments? Are the sheep volatile already been tested? Are sheep or lambs particularly attractive to mosquitoes?

Methodology writing is very confusing in both chemical analysis and behavioral bioassays. Maybe some schematic draw of the devices used for “attractancy assays for thialdine and sheep wool hydrodistillate” could help to understand at least this part.

Results and discussion section need to be better summarized and clarified.

Across the manuscript too many times sentences are mistaken or are unclear, below there are some of them.

L40, what is “CDC trap”?

L41, unclear. I was understanding that only extracts were tested here, not “vibrating wool”

L44-45 this sentence has no meaning here.

L79-80 this sentence has no meaning here, should be contextualized at the beginning.

The first paragraph of MM should be named “chemicals”

L109-111 bad English style.

L122, change in “eluted with 200mictroliers of hexane” and add if was stored or not before the analysis.

L127, “Headspace (HS) collection by solid-phase microextraction” was a black collection with empty vial sealed with parafilm carried out? Why the authors didn’t use a vial with silicon hole cap to carry out this study? Who was the furnisher of the fibers?

L166 define “RRI”. If Kovat index was calculated it should be better mentioned how.

GC-MS analysis; why carry out twice the analysis one in FID and once in GC-MS? Why with different oven temperature programs?

L164-175 this part is very confusing and need rewriting.

L200 “The bioassay of sheep wool hydrodistillate and thialdine…” bad English. L200-205 unclear.

L216, give some reference for this compound as attractant.

L463, why “compounds” in capital letter?

Author Response

Response to Reviewer 4 Comments

Point 1: The manuscript deals with an interesting topic. However, overall it is pretty cumbersome, unclear in methodology and with poor results as also admit the same authors (see lines 365-367).

These aspects make the paper difficult to read and hard to understand.

Response 1: We thank the reviewer for this comment.

  • We separated the results and discussion to improve clarity.

 We have moved to Discussion sections

- “3.1.4. (now 4.1) Comparison of the extraction techniques….”,

- “3.1.5. (now 4.2) Extracted sheep wool volatiles and their …”,

- And last paragraph of “3.2.2. (now in section: 4.3 Bioassays) The attractant activity of sheep…”  

  • We have added several paragraphs to Discussion. Lines 514-566:

In section 4.2: last several paragraphs added (highlighted)

In section 4.3: 1.5 paragraph added.

Point 2: In introduction before the authors claim about their research, they should contextualize it much more in depth. Why they decided to carry out this experiments? Are the sheep volatile already been tested? Are sheep or lambs particularly attractive to mosquitoes?

Response 2: Thank you for this comment

  • Right, there are a few reports that sheep odors can attract mosquitoes [6-8]. Please see from line 72.
  • In response to this comment we have rearranged the sentences and paragraphs in introduction to make the second paragraph about the odors/compounds from sheep that attracted insects according to [6-9].

We have moved the sentence to the second paragraph of introduction

Point 3: Methodology writing is very confusing in both chemical analysis and behavioral bioassays. Maybe some schematic draw of the devices used for “attractancy assays for thialdine and sheep wool hydrodistillate” could help to understand at least this part.

Response 3: We have added two new figures for both methods:

  • 2.1. Hydrodistillation: (Figures 1)
  • 5.1. Glass tube attractancy assayand: (Figure 3).

Point 4: Results and discussion section need to be better summarized and clarified.

Response 4: We have separated the sections, and added more discussions.  

Point 5: Across the manuscript too many times sentences are mistaken or are unclear, below there are some of them.

L40, what is “CDC trap”?

Response 5: We have replaced “CDC” with “U.S. Centers for Disease Control (CDC)” in abstract and in the introduction where first mentioned (changes are highlighted)

Point 6: L41, unclear. I was understanding that only extracts were tested here, not “vibrating wool”

Response 6: in the Abstract after “the extraction techniques or types of fibers applied.” we have added a sentence:

“Two types of bioassays were conducted to study attractancy of the sheep wool volatiles against mosquitoes: laboratory bioassays using glass tubes, and semi-field bioassays using large screened outdoor cages.”

Point 7: L44-45 this sentence has no meaning here.

Response 7: We have removed the sentence from the abstract: “We assume that during vibration the wool fiber intensively releases volatiles that are not released as easily under static conditions.”

Point 8: L79-80 this sentence has no meaning here, should be contextualized at the beginning.

Response 8: We have moved the sentence to the end of the second paragraph in Introduction: ““Yan at al., [9] studied the volatiles from Merino sheep wool samples collected by headspace SPME, and analyzed by GC-MS –electroantennography; the samples containing octanal and nonanal possessed attractiveness to Lucilia cuprina flies.”  

Point 9: The first paragraph of MM should be named “chemicals”

Response 9: The first paragraph is named now: “chemicals” and accordingly, we have changed the numbers of sub-title names in Materials and Methods.

Point 10: L109-111 bad English style.

Response 10: Section: “2.1.2. Dynamic headspace collection (DHS)… “ Lines 141-143:

We replaced: “Sheep wool (100 g) was placed in a polyester Reynolds oven bag (482 mm x 596 mm, Reynolds consumer products, Lake Forest, IL, USA) which prior to use was baked in the oven for 10 h at 80 °C………Prior to the experiment this tube was washed with hexane and after verifying purity of this hexane-wash by GC-MS, this collection tube was baked in the oven at 50-60 °C, for 5 h”

 with 

“Sheep wool (100 g) was placed in a polyester Reynolds oven bag (482 mm x 596 mm, Reynolds consumer products, Lake Forest, IL, USA), which prior to use was baked in the oven for 10 h at 80 °C. ……... Prior to the experiment this collection tube was washed with hexane; purity of this hexane-wash was verified by GC-MS, and the tube was baked in the oven at 50-60 °C, for 5 h..”

Point 11: L122, change in “eluted with 200mictroliers of hexane” and add if was stored or not before the analysis.

Response 11: Section 2.1.2. Dynamic headspace collection (DHS) by Hayesep-Q polymer adsorbent, last sentence: We have revised sentence as suggested. Line 159-160:

“...eluted with 200 μL of hexane into a glass vial, and stored at 4 °C before analysis.”

Point 12: L127, “Headspace (HS) collection by solid-phase microextraction” was a black collection with empty vial sealed with parafilm carried out? Why the authors didn’t use a vial with silicon hole cap to carry out this study? Who was the furnisher of the fibers?

Response 12: Section 2.1.3. Headspace (HS) collection by solid-phase microextraction (HS-SPME): 

  • Yes, we conducted the blank collection with a parafilm and we have added this information into the text.

“The blank collection with the parafilm was performed, as well, at the same conditions.”

  • Parafilm is very practical and we added corresponding citation [21] into the text.

“Sheep wool (0.2 mg) was transferred to a 15 mL vial and sealed with parafilm [21].” Lines 168-172” 

Point 13: L166 define “RRI”. If Kovat index was calculated it should be better mentioned how.

  • Response 13:

Point 14: GC-MS analysis; why carry out twice the analysis one in FID and once in GC-MS? Why with different oven temperature programs?

Response 14: Our results show that chemical compositions detected by these systems for the same extracts are not identical. Combined GC-FID and GC-MS, run under the same temperature program. But the two GC-MS systems were equipped with the different columns that could detect different compounds. The temperature programs were set up to achieve the best resolution and separation.

Lines 471-474: In response to the comment we have added to the text: 

“Although the two methods GC-MS and GC-FID/GC-MS, have advantages and disadvantages [47], both analytical approaches provide important information about the composition of the extracts.

Point 15: L164-175 this part is very confusing and need rewriting.

Response 15: section, 2.2.2. GC-flame ionization detector (FID) and GC-MS with electron impact (EI) ion source

Lines 199-211:

We have revised this part and added Kovats calculation in it. New number for reference [10] is [22]:  

  • Lines 199-202: “The GC oven temperature program was set at an initial temperature of 60 °C and held at that temperature for 10 min post injection, ramped at 4 °C/min to 220 °C, held at that temperature for 10 min, ramped at 1 °C/min to 240 °C.” 
  • Lines 204-208:  Compound Identification: Identification of the volatile components was carried out by comparison of their relative retention times with those of authentic samples or by comparison of their relative retention indices (RRI)/Kovats retention indices, to series of n-alkanes [22]; compounds were analyzed under the same conditions, and compared with literature data.”

Point 16: L200 “The bioassay of sheep wool hydrodistillate and thialdine…” bad English. L200-205 unclear.

Response 16: Section 2.5.1.:

We have revised the sections. Lines 239-251

Point 17: L216, give some reference for this compound as attractant.

Response 17:  Section 2.5.1:  we have added reference:

“The positive control was 1 µg/cm2 of the attractant 1-octen-3-ol [27].” Line 262

Point 18: L463, why “compounds” in capital letter?

Response 18: Line 582 we have replaced the capital letter in “compounds”

Round 2

Reviewer 1 Report

The manuscript can be accepted 

Author Response

Date:           February 12, 2022

To:            Ms. Jessie Jiao, Assistant Editor, MDPI

   MDPI Beijing Office, Tongzhou

E-Mail:    jessie.jiao@mdpi.com

Subject:   Manuscript ID: insects-1575725          

Dear Ms. Jessie Jiao,

Thank you for sending us the additional comments by the reviewers on this manuscript.  Our detailed responses to the reviewer comments please find below. If you need more information, please let me know.

Sincerely,

Maia Tsikolia, PhD

Correspondence author

(The new changes in manuscript are highlighted in blue)

Reviewer #1

Point 1: The research design and the methods description could be improved.

Response 1. We thank the reviewer for reviewing the manuscript. We have made additional changes to improve it.

Reviewer #2

Point 1.Although the author made considerable efforts in their experiment, I continue to consider the study cumbersome and confusing. The results are in part disappointing as no chemicals elicited attraction. Overall the main result obtained is the attraction elicited by the vibrating wool.

Response 1. We thank the reviewer for this comment.

  • The volatiles that did not show attraction were those from hydrodistillate.

  • The vibrating raw sheep wool elicited attraction. We investigated the volatile composition of raw sheep wool using various headspace methods.

  • Section 4.3 Bioassays:To improve this part of discussion we have revised this section.

Point 2. The conclusion is only partially supported by the results. For example, if authors claim that “We assume that under dynamic conditions stored sheep wool more intensively releases volatiles while the emission is limited when wool is not moving” why they didn’t try to collect the VOCs from vibrating wool?

Response 2.

  • Thank you for pointing this out. We plan to extract VOCs from the vibrating wool in the near future.

Point 3. In the conclusion, that is again summarizing the chemical part, I think it would be better to focus on the behavioural results, and how the authors think they can be exploited for future studies or application against mosquitoes, also in consideration that “thialdine didn’t show any significant attractant activity”

Response 3.

  • Conclusion:We have improved the conclusion by adding behavioral results (highlighted).

“In baits with vibrated sheep wool, attraction to mosquitoes could be influenced by combination of olfactory, and visual cues (also, sound from vibrated Vortex might influence mosquito behavior). Of these, olfactory signals seem to be the most important.”

  • 4.3 Bioassays.Last sentence of the paragraph: We have added a sentence to emphasize that thialdine is formed during distillation: “The major component of the hydrodistillate, thialdine, which is product of cooking [45], was not attractive to adult female Ae. Aegipty,according to glass tube bioassays.”

Point 4. About methodology used, the index of attraction used in the study is giving an idea of the insect orientation, however in my opinion, the use of the real captures even log/transformed followed by anova or t-test would give a clear idea of what is happening.

Response 4. Thank toy for this comment.We have used t-test for glass tube assays, and pared t-test for semi-field assays. To make this clear we have rearranged or modified the following paragraphs:

  • InMaterials and methods. In 2.5.2. Semi-field attractancy assays …: Under Figure 5.We have moved:“…the mean number of mosquitoes mean ± SEM (standard error of the mean) for each bait, and the mean of the difference of these means for each pair ± SED (standard error of difference) were used.” Before Before “…Additionally,relative attractiveness (RA) was calculated, similar…”

  • InMaterials and methods. In 2.6. Statistical analysis: We have modified the sentences: A two-tailed Student’s t-test was used to analyze the results for glass tube attractancy assays. A two-tailed paired t-test was used to compare the collections of the pared baits in semi-field attractancy bioassays.

  • InMaterials and methods. In 2.2. The attractant activity of sheep wool using semi-field bioassays:We have moved the paragraph with paired t-test results before table for and added Mean values when presenting the results in the paragraph (highlighted). A paragraph with relative attractiveness results we have moved after Table 4.

Point 5. In the discussion the authors should point out why the results of dymamic hs, SMPE and hydrodistillation were so different. Moreover, if the thialdine was not identified in dynamic headspace or SMPE samples this make me think is not a very volatile compound and this is one of the reason of the lack of insect response.

Response 5.Thank you for this comment.

  • Thialdine could not be identified in the HS methods because it is product of cooking (in our case – hydrodistillation), as described in reference 45. We have added this reference to discussion as well.
  • In Discussion. In 4.1Comparison of the extraction techniques for chemical analysis. A paragraph before the last paragraph. We have modified the paragraph to address the reviewers point.

Point 6. Line 18, which insects? A little too generic

Point 7. Lines 18-22 little confusing, rewrite.

Response 7.We have revised the lines:

Point 8. Line 24, CDC trap?

Response 8.  We have replaced “CDC” with “U.S. Centers for Disease Control (CDC)”

Point 9. Line 38-45, does these data support the conclusion or the objective of the paper? The mosquitoes were attracted by a chemical cue or by a visual one? 

Response 9.  

  • Abstract:We have added a sentence: “Sheep wool, when vibrated, may release intensively volatile compounds, which could serve as olfactory cues, and play significant role in making the bait attractive to mosquitoes.”
  • In section 4.3 Bioassays.We have added 2ndand 3rd
  • In Conclusion:We have added: “In baits with vibrated sheep wool, attraction to mosquitoes could be influenced by combination of olfactory, and visual cues (also, sound from vibrated Vortex might influence mosquito behavior). Of these, olfactory signals seem to be the most important.

Point 10. In introduction and discussion give the order and family of the species mentioned.

Response 10. We have added order and family to the insect name.

Point 11. Line 161, polydimethylsiloxane is misspelled

Response 11. Line 169: we have added the missing letter “p”.

Point 12. Lines 405-406 this goes in MM section

Response 12. We have this sentence and reorganized the statistical section as we have indicated for Point 4.

Point 13. Lines 553/555 this part is redundant.

Response 13. We have removed this part:

 “…identified as active compounds, mostly attractants, against mosquitoes, ranging from ~2-42% of total composition of the extract…”

Point 14. 657-659, why this was not tested or investigated?

Response 14. The lines  657-659 are outside of the text, but regarding the future work we have added:

 “In the future, on the basis of identified potential attractants from the raw sheep wool volatiles, various blends could be tested for attractancy against mosquitoes, to find new active volatile combinations that increase mosquito capture in the traps. Research on raw sheep wool could be extended to test sheep wool baits with CO2, and light; experiments could be conducted with other natural materials.”

Additional changes

  • We have revised figure 1
  • All additional changes are highlighted in blue.

Reviewer 4 Report

Although the author made considerable efforts in their experiment, I continue to consider the study cumbersome and confusing. The results are in part disappointing as no chemicals elicited attraction. Overall the main result obtained is the attraction elicited by the vibrating wool.

The conclusion is only partially supported by the results. For example, if authors claim that “We assume that under dynamic conditions stored sheep wool more intensively releases volatiles while the emission is limited when wool is not moving” why they didn’t try to collect the VOCs from vibrating wool?

In the conclusion, that is again summarizing the chemical part, I think it would be better to focus on the behavioural results, and how the authors think they can be exploited for future studies or application against mosquitoes, also in consideration that “thialdine didn’t show any significant attractant activity”

About methodology used, the index of attraction used in the study is giving an idea of the insect orientation, however in my opinion, the use of the real captures even log/transformed followed by anova or t-test would give a clear idea of what is happening.

In the discussion the authors should point out why the results of dymamic hs, SMPE and hydrodistillation were so different. Moreover, if the thialdine was not identified in dynamic headspace or SMPE samples this make me think is not a very volatile compound and this is one of the reason of the lack of insect response.

Line 18, which insects? A little too generic

Lines 18-22 little confusing, rewrite.

Line 24, CDC trap?

Line 38-45, does these data support the conclusion or the objective of the paper? The mosquitoes were attracted by a chemical cue or by a visual one? 

In introduction and discussion give the order and family of the species mentioned.

Line 161, polydimethylsiloxane is misspelled

Lines 405-406 this goes in MM section

Lines 553/555 this part is redundant.

657-659, why this was not tested or investigated?

Author Response

(The authors gave the same response as above.)

Round 3

Reviewer 4 Report

Overall, the paper has been indeed improved from its last version.